# An Enhanced RIME Optimizer with Horizontal and Vertical Crossover for Discriminating Microseismic and Blasting Signals in Deep Mines

**DOI:** 10.3390/s23218787

**Published:** 2023-10-28

**Authors:** Wei Zhu, Zhihui Li, Ali Asghar Heidari, Shuihua Wang, Huiling Chen, Yudong Zhang

**Affiliations:** 1School of Resources and Safety Engineering, Central South University, Changsha 410083, China; csuzhuwei@csu.edu.cn (W.Z.); zhihuili0807@gmail.com (Z.L.); 2School of Surveying and Geospatial Engineering, College of Engineering, University of Tehran, Tehran 1417466191, Iran; aliasghar68@gmail.com; 3Department of Biological Sciences, Xi’an Jiaotong-Liverpool University, Suzhou 215123, China; shuihuawang@ieee.org; 4School of Computing and Mathematical Sciences, University of Leicester, Leicester LE1 7RH, UK; 5Key Laboratory of Intelligent Informatics for Safety & Emergency of Zhejiang Province, Wenzhou University, Wenzhou 325035, China

**Keywords:** RIME, machine learning, swarm intelligence, feature selection, microseismic, blasting

## Abstract

Real-time monitoring of rock stability during the mining process is critical. This paper first proposed a RIME algorithm (CCRIME) based on vertical and horizontal crossover search strategies to improve the quality of the solutions obtained by the RIME algorithm and further enhance its search capabilities. Then, by constructing a binary version of CCRIME, the key parameters of FKNN were optimized using a binary conversion method. Finally, a discrete CCRIME-based BCCRIME was developed, which uses an S-shaped function transformation approach to address the feature selection issue by converting the search result into a real number that can only be zero or one. The performance of CCRIME was examined in this study from various perspectives, utilizing 30 benchmark functions from IEEE CEC2017. Basic algorithm comparison tests and sophisticated variant algorithm comparison experiments were also carried out. In addition, this paper also used collected microseismic and blasting data for classification prediction to verify the ability of the BCCRIME-FKNN model to process real data. This paper provides new ideas and methods for real-time monitoring of rock mass stability during deep well mineral resource mining.

## 1. Introduction

Many challenges arise as mineral resources are extracted at greater depths, including elevated ground stress, increased well depth, and heightened subsurface temperature, rendering the mining environment increasingly intricate. Engineering disturbances, such as rock drilling and blasting, can influence the stress distribution of deep well rock masses [1]. The rock mass exhibits heterogeneity due to the presence of diverse joints and cracks [2]. When subjected to external pressures, the rock mass experiences a disturbance, resulting in the fracturing of fissures within the rock mass. These fractures subsequently propagate and enlarge until the accumulated energy is released as elastic waves. In instances of significant severity, this phenomenon can result in the destabilization of rock masses, the collapse of regions that have been mined out, the occurrence of roof falls and rock bursts, and several other calamities related to rock masses. Implementing advanced and precise preventative and control methods poses challenges to the safe and effective recovery of deep mineral resources [3]. Hence, the pressing issue at hand pertains to effectively monitoring the real-time stability of rock masses during deep well natural resource mining, appropriately managing areas of pressure concentration, and guaranteeing the safety of humans and equipment.

The primary purpose of microseismic monitoring equipment is to observe and analyze the vibrational signals that arise from the fracturing of rock masses during excavation activities. The stability of the rock mass is evaluated by evaluating and processing the signals, solving the source parameters, and studying the spatiotemporal evolution law of microseismic events. Microseismic monitoring technology’s efficacy is contingent upon the precision of microseismic signal detection [4]. The signals obtained from the existing microseismic monitoring system are frequently contaminated with significant levels of noise, including disturbances caused by blasting and rock drilling activities. This interference significantly hampers the effectiveness of microseismic monitoring technology in analyzing the stability of rock masses and issuing timely warnings for potential disasters. Historically, operators at the site have relied on their personal expertise to categorize microseismic and blasting data, a process that has been both time-consuming and demanding in terms of labor. Moreover, this approach has been prone to subjective interpretation. Consequently, to address this issue, a growing body of scholars has been investigating it.

Although previous studies [5,6] have achieved outstanding results in the field of microseismic and blasting evaluation, most of them were still insufficient in efficiently identifying the two types of signals due to the complex on-site production environment, multiple and mixed noise sources, and large blasting impact. This resulted in a lot of background noise and useless data mixed with adequate microseismic information. In addition, there was a large overlap between the frequency distribution and other source parameters of microseismic and blasting events. In order to address this issue, this paper attempted to develop a model with strong identification capabilities based on fuzzy k-nearest neighbor (FKNN). FKNN is an excellent classification model with no prior data distribution assumptions, no training needs, and simple thinking. As a result, it is widely used and has a wide range of applications. There have also been many related studies in recent years.

Zhao et al. [7] introduced the DPC-FWSN algorithm, which uses fuzzy and weighted shared neighbor methods for clustering uneven density datasets. By employing an improved binary salp swarm algorithm-based FKNN approach (IBSSA-FKNN) to differentiate between patients with mild cognitive impairment, Alzheimer’s disease, and healthy controls, Lu et al. [8] created an early diagnostic method for Alzheimer’s disease (AD). To diagnose epileptic EEGs, Liu et al. [9] suggested a modified binary grey wolf optimization-based fuzzy KNN method. A GSHHO-FKNN machine learning model was created by Zhang et al. [10] by combining an enhanced Harris hawk’s optimization (GSHHO) based on the Gaussian mutation mechanism and a simulated annealing approach with a fuzzy KNN. Using a machine learning technique that combines an enhanced binary mutant quantum grey wolf optimizer (MQGWO) with FKNN, Hu et al. [11] examined 3069 data points from 314 HD patients. Ahmad et al. [9] offered an ensemble classifier composed of FKNN, random forest, k-nearest neighbor, and support vector machine to enhance prediction outcomes further. In order to concurrently execute feature selection and parameter optimization, Wu et al.’s [12] enhanced sine cosine algorithm-based optimization approach was used to create an FKNN model. In order to generate the membership matrix, Wan et al. [13] used FKNN to combine the intraclass and interclass weighted matrices.

However, two important parameters affect FKNN’s classification accuracy and are also extremely important for feature extraction of microseismic and blasting images in deep areas. Therefore, optimizing FKNN’s parameters is key to establishing strong identification capabilities. Parameter optimization methods have unique advantages in finding the best parameters of a classifier or regressor, or solutions for a certain goal selected from many sub-optimal solutions [14,15,16]. By monitoring and mimicking the behavioral traits of natural occurrences like biological foraging and nesting, swarm intelligence optimization approaches have offered several innovative and effective optimization strategies. They provide fresh approaches and methods for resolving several complicated issues and have high parallelism and autonomous exploration capabilities. They exhibit remarkable performance across various domains such as feature selection [17,18], image segmentation [19,20], economic scheduling [21,22], multi-objective problems [23,24], expensive optimization problems [25,26], multi-attribute decision making [27,28,29,30], and disease diagnosis [31,32].

Numerous swarm intelligence optimization techniques have been presented in recent years, such as the RIME [33], multi-verse optimizer (MVO) [34], bat-inspired algorithm (BA) [35], Harris hawks optimization (HHO) [36], particle swarm optimization (PSO) [37], salp swarm algorithm (SSA) [38], whale optimization algorithm (WOA) [39], weighted mean of vectors (INFO) [40], Runge Kutta optimizer (RUN) [41], colony predation algorithm (CPA) [42], stochastic fractal search (SFS) [43], slime mold algorithm (SMA) [44,45], and hunger games search (HGS) [46]. In addition, in order to further improve the performance of swarm intelligence optimization algorithms, many variant algorithms have been proposed, such as the bat algorithm based on collaborative and dynamic learning (CDLOBA) [47], the ant colony optimizer with grade-based search (GACO) [48], hybridizing grey wolf optimization with differential evolution (HGWO) [49], evolutionary biogeography-based whale optimization (EWOA) [50], the chaotic mutative moth-flame-inspired optimizer (CLSGMFO) [51], the moth-flame optimizer with mutation strategy (LGCMFO) [52], Cauchy and Gaussian sine cosine optimization (CGSCA) [53], and the double adaptive random spare reinforced whale optimization algorithm (RDWOA) [54], the A–C parametric whale optimization algorithm (ACWOA) [55], Harris hawks optimization with Gaussian mutation (GCHHO) [56], an enhanced sine cosine algorithm (LSCA) [57], and the Gaussian kernel probability-driven slime mold algorithm with new movement mechanism (MGSMA) [58]. Among them, RIME is a recently proposed optimization algorithm with strong optimization performance [33]. Therefore, this paper proposes a RIME based on vertical and horizontal cross-cross search through the study of RIME to enhance the performance of FKNN and improve the identification ability of microseismic and blasting.

The main contributions and innovations of this paper are as follows:Through examining and analyzing the RIME algorithm, this paper proposes a novel method called CCRIME, which is based on vertical and horizontal crossover. The introduction of CCRIME not only enhances the quality of the solutions found but also improves the overall search capabilities.To enhance the classification capabilities of the FKNN model, a binary version of CCRIME was created through the use of the binary transformation method. This approach aimed to optimize the important parameters inside the FKNN model. The CCRIME-FKNN model, which is optimized for CCRIME, is an abbreviation for the CCRIME-optimized FKNN model.The performance of CCRIME, an optimization method based on swarm intelligence, was evaluated using 30 benchmark functions from IEEE CEC2017. The results of this study demonstrated that CCRIME exhibits exceptional performance across several perspectives, establishing it as a highly effective algorithm.This study utilized microseismic and blasting images to extract and select appropriate features. Through the application of CCRIME-FKNN, the identification of microseismic and blasting events was successfully achieved, resulting in a high level of accuracy.

The remaining content of this paper is arranged as follows: Section 2 describes the collected microseismic and blasting data. Section 3 elaborates on the details of the proposed CCRIME. Section 4 explains how to use CCRIME to optimize FKNN. Section 5 analyzes the performance of CCRIME using benchmark functions and identifies microseismic and blasting signals using CCRIME-FKNN. Section 6 summarizes the research work carried out in this paper and points out future research directions.

## 2. Related Work

### 2.1. Data Description

Linglong Gold Mine is one of the gold mines with the longest mining history and the deepest mining depth in China. The surface elevation of the Daikoutou mining section in the Jiuqu mining area is +255, and the current mining depth has reached below 900 m underground. Due to the simple, early mining methods and the failure to fill the mined-out areas, a large number of mined-out areas have been left under the ground. Within an area affected by a mined-out area, secondary stress is highly concentrated, and the surrounding rock of the ore body is granite with a strong tendency toward rock burst. When carrying out tunnel excavation and other production activities in this area, blasting disturbance causes stress redistribution, quickly forming stress concentration areas and thus triggering geological disasters such as rock bursts. Significant rock burst phenomena have occurred during the development of the −620 m, −670 m, and −694 m levels, causing injuries to underground personnel and loss of mine property.

As the mining depth continues to increase, the ground stress also increases, and the risk of geological disasters such as rock bursts gradually increases. To this end, Linglong Gold Mine has carried out scientific research in cooperation with Central South University to build an online microseismic monitoring system for ground pressure in the Daikoutou mining area for real-time and long-term monitoring of deep areas. The waveform information collected by the IMS microseismic monitoring system automatically picks up the arrival time of P-waves and S-waves through its own software, TRACE. The software lists the number, code, and order of sensors triggered by microseismic monitoring signals, displays the moment when signals trigger the sensors, and collects waveform images from each sensor, as shown in Figure 1.

The monitoring technician screens the blasts and microseismic events by comparing the source trigger time with the source waveforms. Accurate source localization is ensured by calibrating the P-wave arrival time based on the waveform onset time. Due to the difference in the generation mechanisms of the blasts and microseismic events, the two waveforms have their own characteristics. For example, the energy of the blast waveform is concentrated in the front end of the rapid decay without an obvious S-wave, and the energy of the microseismic events waveforms is evenly distributed in the window with an obvious S-wave. The amplitudes of the blast waveforms are generally larger than that of the microseismic event waveforms. The microseismic monitoring database of the Linglong Gold Mine was constructed from the artificially screened blasts and microseismic events waveforms to analyze the microseismic activities and evaluate the engineering health of the rock mass. Therefore, the original data source for this study is the waveform images collected in the Linglong Gold Mine microseismic monitoring system database.

### 2.2. Microseismicity and Blasting

Since microseismicity and blasting have different characteristics and effects, understanding the differences between them can help us better understand and respond to earthquakes and blasting activities. For example, recognizing microseismic activity can better predict possible large earthquakes and facilitate appropriate preventive measures. As for blasting, we can minimize the negative impact on the environment by understanding its characteristics and impacts and taking the necessary measures. Computer-aided technology can improve the accuracy of identifying microseisms and blasting by automating the analysis of large amounts of data. Compared with traditional manual identification methods, computer-aided technology can eliminate the influence of human factors and reduce the possibility of misjudgment and omission.

For example, Jiang et al. [59] suggested a time-frequency discrimination approach to automatically detect these signals and produce more trustworthy data for mass spectrometry monitoring analysis. Singular value decomposition is used to decrease the amount of data while obtaining distinctive characteristics. Random Forest (RF) is employed to accomplish automated identification, and the essential characteristics are determined based on changes in the classification error and Gini coefficient. Dai et al. [60] developed a technique for automatically recognizing MS signals in the presence of blasting signal interference. Improved adaptive noise with full integrated empirical mode decomposition, singular value decomposition, and the k-nearest neighbor algorithm are all part of this technique. The original multi-frequency signal is decomposed into many single-frequency signal subcomponents using integrated empirical mode decomposition, and singular values are extracted from the matrix generated by the decomposition results using SVD. 

In order to distinguish between coal mine microseismic signals and blasting signals, Li et al. [61] studied the differences between them by analyzing their main frequency, signal duration, waveform after peak attenuation, and multiple fractal parameters, laying a foundation for automatic identification of mine-used MS signals and blasting signals recorded underground in coal mines. Li et al. [62] analyzed the fractal characteristics of microseismic waves and blasting waves in a certain mine based on the nonlinear characteristics of waveforms using simple fractal and multiple fractal theory. This method can more clearly and significantly identify mine MS and blasting waves, thereby monitoring coal rock fracture more accurately. Peng et al. [63] proposed an automatic classification method for microseismic signals based on the Gaussian mixture model-hidden Markov model. This method only uses the signal’s extracted Mel frequency cepstral coefficient (MFCC) features for training and testing. The results show that the classification accuracy of this method reaches 92.46%, verifying its effectiveness in the automatic classification of underground microseismic data.

## 3. The Proposed CCRIME

### 3.1. An Overview of RIME

RIME is a new and efficient optimization algorithm proposed by Su et al. [33]. It mainly simulates the peculiar physical phenomenon of uncondensed water vapor in the air condensing on branches or objects and freezing when encountering low temperatures. The RIME algorithm consists of 4 main stages: the initial rime ice crystal cluster formation, soft RIME search strategy, hard RIME puncture mechanism, and greedy selection mechanism for the optimal solution.

In the initial stage of rime formation, each search agent in the population is treated as a RIME agent of the RIME algorithm. All RIME agents are combined into a whole as the initial population of the algorithm. First, each agent in the population is initialized within a given search space through Equation (1):(1)$$R_{i}=B_{min}+r_{i}\times (B_{max}-B_{min})$$
where $R_{i}$ represents each search agent, $B_{max}$ and $B_{min}$ represent the upper and lower boundaries, respectively, and $r_{i}$ is a pseudo-random sequence between [0, 1].

The soft RIME search strategy mainly simulates that when free particles move near the soft rime, they will be captured and condensed by the particles in the soft rime and change the stability of the soft rime. When the distance between the particles and the soft rime particles is far, they will not be captured and continue to be in a free state. As free particles are gradually captured and condensed by the soft rime, the coverage area of the soft rime will gradually increase, and its probability of capturing particles will also increase. However, due to environmental factors, the coverage area of the soft rime will not increase indefinitely and will gradually show a steady state. By simulating the movement rules of free particles and soft rime, the specific mathematical model is as follows:(2)$${R_{ij}^{new}=R}_{best,j}+Rf\times \left(h\times \left(B_{\max(i,j)}-B_{\min(i,j)}\right)+B_{\min(i,j)}\right),r_{2}<E$$
(3)$$Rf=r_{1}\times \cos\theta \times \beta$$
(4)$$\;\theta =\pi \cdot \frac{t}{10\cdot T}$$
(5)$$\beta \;=1-\left[\frac{w\cdot t}{T}\right]/w$$
(6)$$E\;=\sqrt{\left(t/T\right)}$$
where $R_{ij}^{new}$ is the position of the free particle after moving and $R_{best,j}$ is the best RIME agent in the RIME population. $r_{1}$ is a random number between [−1, 1]. cos⁡θ will change as the algorithm iterates. β represents the environmental factors in rime and changes as the algorithm iterates, controlling the algorithm’s convergence. h is a random value between [0, 1], simulating the distance between free particles. t and T represent the current number of iterations of the algorithm and the upper limit of the number of iterations, respectively. w controls the number of segments of the step function, with a default value of 5. E represents the probability of free particles being captured. It will change as the algorithm iterates and generally shows an increasing trend.

The hard RIME puncture mechanism mainly simulates that in a stronger wind environment, the movement of particles will be affected by the wind direction, and due to the same growth direction, it is easy for a cross phenomenon called hard rime puncture to occur between each hard rime. In addition, as the hard rime grows, the phenomenon of hard rime puncture will become more frequent. Therefore, the hard rime puncture mechanism can be used for position updates between rime agents, allowing rime agents to exchange particles and improve the convergence ability of the algorithm to help jump out of local optima. The mathematical model of the hard RIME puncture strategy is as follows:(7)$${R_{ij}^{new}=R}_{best,j},r_{3}<F^{normr}(S_{i})$$
where $R_{ij}^{new}$ is the *jth* particle of the best rime agent in the population and $R_{best,j}$ new is the updated particle location. The fitness value of the current agent is normalized to produce $F^{normr}(S_{i})$, which represents the likelihood that the agent will experience a hard rime puncture. A number at random between [0, 1] makes up $r_{3}$.

### 3.2. Horizontal and Vertical Crossover Search

In order to perform the horizontal crossing operation, two different rime agents are generally used. The ability of distinct agents to communicate with and learn from one another as a consequence of the horizontal crossover search significantly increases the agents’ capacity for exploration and accelerates algorithm convergence. Assume that the parent agents $x_{i}$ and $x_{j}$ experience a horizontal crossover operation, which may be described by Equations (8) and (9):(8)$${MS}_{i}^{n}=\varepsilon_{1}\times x_{in}+\left(1-\varepsilon_{1}\right)\times x_{jn}+c_{1}\times \left(x_{in}-x_{jn}\right)$$
(9)$${MS}_{j}^{n}=\varepsilon_{2}\times x_{jn}+\left(1-\varepsilon_{2}\right)\times x_{in}+c_{2}\times \left(x_{jn}-x_{in}\right)$$
where the given equation involves the use of random variables $\varepsilon_{1}$ and $\varepsilon_{2}$, which are uniformly distributed within the interval [0, 1]. Additionally, the variables $c_{1}$ and $c_{2}$ are also uniformly distributed within the interval [−1, 1]. The variables $x_{in}$ and $x_{jn}$ represent the nth dimension of the *ith* and *jth* agents, respectively. The offspring of the *nth* position vectors, denoted by ${MS}_{i}^{n}$ and ${MS}_{j}^{n}$, are generated through the horizontal crisscross operation of the agents $x_{i}$ and $x_{j}$.

The vertical crossover operator is a computational procedure that involves the manipulation of two discrete position vectors belonging to individual agents within a given population. This approach potentially enables certain position vectors that are confined to a local optimum to persist in their search while simultaneously minimizing alterations to the position vectors that are typically explored. Consequently, in the latter phases of the exploration process, ants frequently encounter a local optimum due to the stagnation of specific position vectors. Implementing a vertical crisscross search technique can facilitate inter-agent learning of position vectors, thereby enhancing their capacity to evade a local optimum. Assuming the vertical crossover operation is performed on the *mth* and *nth* position vectors of agent i, as described in Equation (10):(10)$${MS}_{i}^{m}=\varepsilon \times x_{im}+(1-\varepsilon )\times x_{in}$$
where ε represents a random number between [0, 1] and ${MS}_{i}^{m}$ represents the *mth* position vector of the offspring generated by the vertical crisscross operation between the *mth* agent *nth* position vectors of agent i.

### 3.3. The Proposed CCRIME

Since the original research on RIME was only an optimization algorithm proposed by simulating the changing shape of rime, no further research or analysis was conducted on the algorithm’s performance. Further, in this paper, we have found that it has a greater advantage in convergence through benchmark function experiments, and it has further room for improvement. In addition, when solving the feature selection problem, through experiments, we also have found that it has a greater advantage than the general traditional algorithms in finding high-quality individuals for solving practical problems. Therefore, in order to further study the performance of the RIME algorithm and improve its performance, this paper compared RIME with some similar algorithms from multiple angles and found that there is still room for improvement in the quality of the solutions obtained by RIME and its search capabilities need to be further enhanced to avoid falling into local optima.

Based on these facts, this paper found through the study of vertical and horizontal cross-search strategies that it has a high ability to improve the performance of algorithms, and some studies have also used this method to enhance the performance of some optimization algorithms [40,41,42]. Inspired by this, in order to improve RIME’s search capabilities, enhance the quality of the solutions obtained, and avoid falling into local optima during the search process, this paper enhanced the original RIME by using horizontal and vertical crossover search strategies. In the enhancement process, the horizontal and vertical crossover search strategies mainly act after the hard RIME puncture mechanism. By improving the search capabilities of RIME, CCRIME can obtain higher-quality solutions and effectively improve its optimization capabilities in practical applications. The pseudocode is shown in Algorithm 1.**Algorithm 1** Pseudocode of CCRIMEInitialize the population of rime R
Calculate the fitness of each agentSelect the optimal agent**While** t < T
 Update the particle capture probability E
 **If** $r_{2}<E$
  Update rime agent location by Equation (10) **End If** **If** $r_{3}<F^{normr}(S_{i})$
  Cross-updating between agents by Equation (14) **End If**
 **If** $f\left(R_{i}^{new}\right)<f(R_{i})$
  Replace $R_{i}$ with $R_{i}^{new}$
  **If** $f\left(R_{i}^{new}\right)<f(R_{best})$
   Replace $R_{best}$ with $R_{i}^{new}$
  End If **End If**
 Perform horizontal crossover search and vertical crossover search t=t+1
 **End While**

## 4. The Proposed CCRIME-FKNN Model

### 4.1. Binary Transformation Method

The goal of feature selection is to select sample features with obvious distinctiveness, and feature selection based on swarm intelligence algorithms selects the main attributes representing the dataset. Suppose the given feature set is F, represented as $F_{i}=\left\{f_{i,1},f_{i,2},\cdots ,f_{i,j},\cdots ,f_{i,t}\right\}$, where t represents the number of features after preprocessing and i represents the original number of datasets. FS is the new feature subset selected by the feature selection algorithm, represented as $FS_{i}=\left\{fs_{i,1}\right.$, $\left.fs_{i,2},fs_{i,j},\cdots ,fs_{i,m}\right\}$, where m is the length of the selected features and $fs_{i,j}\in \left\{1,0\right\},j=1,2,\cdots ,m$. If $fs_{i,j}=1$, this indicates that the selected feature j in the dataset is a high-quality feature. If $fs_{i,j}=0$, this indicates that the selected feature j in the dataset is useless. In the process of using a binary RIME algorithm to select features, each position of a rime individual represents a subset of a feature dataset. The rime population includes a set of positions of rime individuals, where the position set is represented by binary 0 or 1. The binary representation of the solution space is shown in Equation (11):(11)$$X_{d}(t+1)=\left\{\begin{array}{cc} 1, & \;\mathrm{sigmoid}\;\left(X_{d}(t)\right)\geq \;\mathrm{rand}\;\\ 0, & \;\mathrm{otherwise}\; \end{array}\right.$$

The variable rand denotes a uniformly distributed random number within the interval [0, 1]. The notation $X_{d}(t+1)$ refers to the updated binary solution at time t+1. The mathematical expression for the Sigmoid function is presented in Equation (12) herein, where the variable x represents the solution produced during the CCRIME iteration.
(12)$$sigmoid(x)=\frac{1}{1+e^{-2x}}$$

### 4.2. Feature Extraction Method

The process of obtaining a “non-image” representation or description of an image, such as numerical values, vectors, and symbols, in order to derive useful data or information, is referred to as feature extraction. The resulting features are the extracted “non-image” representations or descriptions. The present study employed color moments, gray-level co-occurrence matrices, and Local Binary Patterns (LBP) as feature extraction techniques. Subsequently, we derived distinctive attributes from the intrinsic shape properties of microseismic and blasting images. These attributes encompass the area, bounding box, major axis, minor axis, eccentricity, orientation, Euler number, equivalent circle diameter, fullness, expansion degree, and perimeter.

The mathematical theory of color matrixes is that all colors in an image can be described using matrix methods. At the same time, because the color distribution of the image is concentrated in low-order matrixes, the distribution of image colors can be described by first-order and second-order colors. This is most effective when it only contains a single target picture. The retrieval results are very effective when using three color coordinate axes, each including color moments axes. In this paper, we used color moments to extract 3 grayscale color features from the microseismic and blasting images, 12 RGB (RGB stands for red, green, and blue channels) color features, and 12 HSI (perceived colors based on three basic characteristics: hue, saturation, and brightness) color features.

As per previous works, the gray-level co-occurrence matrix stands as one of the well-established approaches to statistical texture feature extraction from images. This method mainly obtains features such as energy and inertia of the image gray-level co-occurrence matrix. In terms of statistics, one commonly used method is to use the autocorrelation function of the image to extract texture features of the image. By processing the energy spectrum function of the image, a good extraction of related texture parameters can be achieved. In this paper, for texture features, we extracted 20 gray-level co-occurrence matrix texture features, 6 gray-level co-occurrence smooth matrix texture features, and 15 gray-level co-occurrence gradient matrix texture features.

The LBP operator recognizes the central pixel based on the combined difference distribution between adjacent pixels and the central one. Essentially, it calculates the first derivative of the binary gradient direction between neighboring pixels on a circle and center pixels. Therefore, it also calculates the gradient relationship in local neighborhoods of images. The encoding outcome from the LBP operator reveals certain discernible local attributes in the original images, such as boundaries, corners, flat areas, and feature points. Additionally, histogram statistics can establish a somewhat unique vector to express the image features.

In this paper, for local texture features, we used the basic LBP equivalent LBP rotation invariant to extract 36 features.

### 4.3. Fuzzy k-Nearest Neighbor

The k-nearest neighbor (KNN) algorithm functions as a classification method influenced by neighboring elements. This straightforward, efficient, and non-parametric form of supervised learning hinges on providing a training set equipped with class labels or numerical attributes. For a sample to be classified or predicted, find k nearest training samples to it and then determine its output based on the neighbors’ information. The output is the most common category among neighbors.

The incorporation of fuzzy logic into the KNN algorithm has been proposed to enhance the flexibility and interpretability of the model in the context of classification tasks. This has led to the development of FKNN. In contrast to a solitary KNN class, the process of conducting FKNN classification involves determining the appropriate values for the number of neighbors (k) and the fuzzy strength coefficient (m), as well as selecting an optimization strategy to enhance the k and m values. These factors are crucial in influencing the outcomes of the classification. Consequently, various optimization algorithms can be employed in experimental settings to optimize diverse k and m values and attain the optimal solution for k and m. The formula for determining the degree of membership of data in a fuzzy set is expressed as follows:(13)$$u_{i}(x)=\frac{\sum_{j=1}^{k}\;\;u_{ij}\left(1/{||x-x_{j}||}^{2/(m-1)}\right)}{\sum_{j=1}^{k}\;\;\left(1/{||x-x_{j}||}^{2/(m-1)}\right)}$$
where $u_{i}(x)$ represents the specific membership of vector x. i∈[1:C] represents the number of classes. j∈[1:k] represents the number of nearest neighbors. In addition, the fuzzy strength m parameter is used to determine the distance weight. m represents the degree of distance weighting when calculating the contribution of each neighbor to the membership value.

### 4.4. The Proposed CCRIME-FKNN Model

Feature selection technology is a binary optimization approach, whereas the CCRIME algorithm is specifically developed to address real continuous optimization problems. Consequently, it is not feasible to employ the CCRIME algorithm for directly resolving feature selection issues. During the optimization process, executing a binary operator is necessary to convert real values to binary values. The present study introduces a discrete iteration of CCRIME, namely BCCRIME, which was derived from the continuous CCRIME. The BCCRIME methodology establishes a bounded search range for the solution, specifically between 0 and 1. This approach treats the feature selection problem as a constrained optimization problem. Subsequently, the outcome of the search process is subjected to a Sigmoid function transformation technique, resulting in a binary numerical representation consisting solely of 0 and 1. In this representation, a value of 1 indicates that the model has selected the corresponding feature, while a value of 0 indicates that it has not been selected. A comprehensive flowchart is presented herein (Figure 2).

The evaluation function of the combined use of BCCRIME and the FKNN classifier is deemed overly simplistic and lacks universality. When utilizing the CCRIME algorithm for feature selection, it becomes imperative to reevaluate the approach to establishing the evaluation function. As a preliminary step in data analysis, feature selection necessitates two primary prerequisites. The feature subset acquired exhibits a notable degree of classification accuracy, indicating a robust correlation between the feature subset and the respective category. It is advisable to minimize the number of features in the selected feature subset to mitigate the effects of the curse of dimensionality. Hence, the present study redefined an evaluation function that is appropriate for the BCCRIME-FKNN hybrid model, as depicted in Equation (14):(14)$$F\mathrm{itness}=\alpha \cdot \;\mathrm{error}\;+\beta \cdot \frac{\left|R\right|}{\left|D\right|}$$
where the formula provided denotes the relationship between the error rate of a classifier model, the dimensionality of a dataset, and the number of attributes in the resulting subset. Specifically, the variable error represents the error rate of the classifier model, while the variable |D| represents the dimensionality of the dataset, which refers to the total number of attributes. Lastly, the variable |R| represents the number of attributes in the resulting subset. The variable α is utilized as a metric for assessing the significance of the classification error rate, whereas β denotes the extent of the chosen characteristics. Consistent with prior literature, the present study employs α=0.99 and β=0.01.

## 5. Experiments, Results, and Analysis

This section primarily focused on two aspects and utilized experimental validation to implement the proposed methodology. Initially, a set of benchmark function experiments were carried out to authenticate the efficacy of CCRIME. Subsequently, the CCRIME-FKNN approach was utilized in various feature selection-related classification prediction tasks, successfully showcasing its robust classification prediction abilities. The CCRIME-FKNN method was effectively utilized in the investigation of deep mine microseismic and blasting identification, resulting in outstanding outcomes.

### 5.1. Benchmark Function Validation

In this section, based on the IEEE CEC2017 benchmark function experiment, not only were CCRIME and 11 basic algorithms compared experimentally, but CCRIME was also compared experimentally with 11 advanced variant algorithms, fully demonstrating the core advantages of CCRIME’s strong convergence performance and resistance to falling into local optima.

#### 5.1.1. Experimental Setup

Benchmark datasets are a commonly used tool for evaluating various computational characteristics, using a set of recognized criteria to determine which technology performs best overall [64,65]. The experimental foundation for the benchmark function experiment consisted of 30 benchmark functions from IEEE CEC2017, as presented in Table 1. Throughout the experiment, the primary focus was on comparing basic and advanced variant algorithms. The experiment on algorithm comparison involved several algorithms, namely RIME [33], MVO [34], BA [35], HHO [36], PSO [37], SSA [38], WOA [39], JAYA [66], PO [67], SFS [43], SMA [45], and HGS [46]. The superior performance of the subject has been adequately demonstrated in prior original research, thereby rendering the comparative outcomes with said research compelling. As RIME is categorized as a variant algorithm, it is imperative to conduct a comparative analysis with other advanced variant algorithms such as CDLOBA [47], GACO [48], HGWO [49], EWOA [50], CLSGMFO [51], LGCMFO [52], CGSCA [53], RDWOA [54], ACWOA [55], GCHHO [56], LSCA [57], and MGSMA [58]. The variant algorithms that are involved in the participation not only exhibit commendable performance but have also demonstrated successful application across various domains. In order to guarantee equity and precision in the experimental procedure, all algorithms utilized in the comparison were executed under identical circumstances. This study established a population size of 30 and uniformly designated a maximum evaluation number of 300,000 iterations. Furthermore, the algorithms underwent 30 independent tests in order to mitigate the influence of stochastic variables. Comprehensive statistical analyses were conducted on the experimental outcomes of the benchmark functions, utilizing measures such as the mean, variance, Wilcoxon signed-rank test, and Friedman test.

#### 5.1.2. Comparison with Basic Algorithms

This section presents a comparison between CCRIME and 11 fundamental algorithms in IEEE CEC2017. This study conducted a comparison of various algorithms, namely RIME [33], MVO [34], BA [35], HHO [36], PSO [37], SSA [38], WOA [39], JAYA [66], PO [67], SFS [43], SMA [45], and HGS [46]. Table A1 displays the statistical measures of the central tendency and variability, specifically the mean and standard deviation, respectively. The “AVG” column denotes the arithmetic mean, while the “STD” column represents the standard deviation. These values were obtained as a result of the conducted experiment. Upon examination of the mean and standard deviation, it has been determined that CCRIME achieved the lowest mean on 21 functions, while SSA achieved the lowest mean on 3 functions. Additionally, RIME, BA, and PSO each achieved the lowest mean on 2 functions. An analysis of the mean revealed that CCRIME exhibited superior performance on over two-thirds of the functions, thus implying its robust optimization capabilities and ability to yield solutions of high quality. Furthermore, it is evident that CCRIME exhibited favorable performance with respect to the standard deviation, implying its commendable stability.

Moreover, Table 2 presents the performance hierarchy of all algorithms across individual benchmark functions, as well as the comprehensive ranking across 30 benchmark functions. Through empirical observation, it is evident that CCRIME achieved the highest ranking in all categories of functions within IEEE CEC2017, as well as securing the top position in the overall ranking across 30 functions. This serves as a comprehensive demonstration of the exceptional performance of CCRIME. Furthermore, for the purpose of enhancing the dependability of the experimental outcomes, Table 2 furnishes the analysis outcomes of the Wilcoxon signed-rank test. The symbols “+” and “−” are employed to denote that CCRIME outperformed and underperformed the comparison algorithm, respectively, while “=” signifies that CCRIME and the comparison algorithm exhibited comparable performance. The analysis results indicated that CCRIME outperformed other algorithms on 21 out of 30 functions, thereby providing substantial evidence to support the superior performance of CCRIME.

Following the Wilcoxon signed-rank test analysis, the ranking results subsequent to the Friedman test analysis are presented in Figure 3. These results indicate that CCRIME holds the top rank with a score of 2.55, followed by RIME at No. 2 with a score of 4.22, and MVO at No.3 with a score of 4.65. Consequently, upon conducting an analysis, it can be deduced that CCRIME exhibited a noteworthy edge over RIME, which is ranked second, and MVO, which is ranked third. This observation further implies that CCRIME holds a superior position in comparison to other algorithms. Figure 4 illustrates the convergence behavior of various functions to showcase the efficacy of CCRIME. The results indicate that CCRIME outperformed other fundamental algorithms in terms of convergence, and it exhibited robust search capabilities.

Consequently, the experimental analysis presented above provides evidence that CCRIME exhibited strong convergence performance and a robust ability to avoid local optima in the basic algorithm comparison experiment.

#### 5.1.3. Comparison with State-of-the-Art Variants

This section presents a comparative analysis between CCRIME and 11 advanced variant algorithms, whereby CCRIME is a derivative algorithm that incorporates the horizontal and vertical crossover strategy into RIME. Table A2 presents the mean and standard deviation values for each function. The results indicate that CCRIME achieved the highest mean value across 17 functions, while CDLOBA and LSCA obtained the highest mean value across 3 functions. Additionally, GCHHO, LGCMFO, and MGSMA demonstrated strong performance on 2 functions, while GACO performed well on 1 function. An analysis of the average performance indicates that CCRIME exhibited a distinct superiority in comparison to alternative algorithms. Furthermore, the performance of CCRIME in relation to sexually transmitted diseases is noteworthy, indicating a level of stability.

Table 3 presents the ranking of all algorithms on each of the 30 benchmark functions, with the aim of conducting a more in-depth analysis of CCRIME’s performance. The results demonstrate CCRIME’s clear advantage over other comparable algorithms, both in terms of its performance on individual functions and its overall performance. The findings of the Wilcoxon signed-rank test confirm the superior performance of CCRIME. Specifically, CCRIME outperformed other comparable algorithms on 20 benchmark functions.

Subsequently, the Friedman test was employed to evaluate the efficacy of the algorithms. The ranking results of each algorithm are presented in Figure 5 according to the Friedman test. Upon examination of the ranking outcomes, it is evident that CCRIME attained the topmost position with a score of 2.59, while MGSMA secured the second position with a score of 4.10. This outcome effectively signifies that CCRIME outperformed MGSMA and establishes its superiority over other analogous algorithms. Figure 6 displays the convergence curves of various algorithms on certain functions. It is evident that CCRIME outperformed other comparable variant algorithms in terms of convergence performance.

The core performance of CCRIME, an optimization algorithm based on swarm intelligence, was demonstrated through a comparative experiment with advanced variant algorithms. The results effectively indicate that CCRIME is an outstanding algorithm in this field. Owing to its formidable optimization capabilities, the proposed CCRIME exhibits versatility suitable for various applications, including but not limited to the resolution of the blocking flowtops problem [68], object tracking [69,70], plant disease recognition [71], and constrained optimization [72,73,74].

### 5.2. Feature Selection Experiments

This section first describes the evaluation criteria for the earthquake data prediction experiment. Then, the proposed BCCRIME-FKNN model is compared with other similar classic prediction models, such as BP, CART, RandomF, etc. Further, BCCRIME-FKNN is compared with other BCCRIME-based models, such as KNN, KELM, MLP, etc. Next, BCCRIME-FKNN is compared with other swarm intelligence-based FKNN prediction models, such as BPSO, BCS, etc. Finally, the key features selected by BCCRIME-FKNN are shown.

#### 5.2.1. Experimental Setup

Since the original source of raw data for this paper is the waveform images acquired by each sensor, color moments, gray-level covariance matrices, and local binary patterns (LBP) were mainly used as feature extraction techniques in the preprocessing stage of the data. Subsequently, we extracted some shape attributes from the microseismic and blast images, including the area, bounding box, major axis, minor axis, eccentricity, orientation, Euler number, equivalent circle diameter, fullness, expansion, and perimeter. The experiments mainly used the ten-fold cross-validation approach to train, test, and validate the proposed algorithms. The experiment on feature selection was conducted using the benchmark function’s experimental environment, focusing on incorporating five distinct feature selection evaluation metrics. The content in question is as follows.

A confusion matrix is a visual representation that documents the outcomes of classification predictions within the pattern recognition domain. The assessment of a classification models’ performance is a crucial task, and a key indicator of this is the correlation between the predicted class and the true class attributes of the sample data. A confusion matrix can be utilized to derive evaluation metrics such as accuracy, specificity, and sensitivity. Binary classification problems involve categorizing samples into two distinct groups, namely positive and negative. The classification evaluation tool known as the confusion matrix comprises four distinct categories: true positives (*TP*), true negatives (*TN*), false positives (*FP*), and false negatives (*FN*).

The Accuracy (ACC) metric is a widely employed classification evaluation measure that assesses a classifier’s capacity to accurately recognize samples. The accuracy range is bounded by the interval [0, 1]. A higher accuracy value, closer to 1, indicates superior classification performance of the classifier. The methodology for computing accuracy is as follows:(15)$$Accuracy=\frac{TP+TN}{TP+FP+FN+TN}$$

The term “specificity” (SPE) pertains to the likelihood of correctly identifying an individual who is truly negative as negative. The metric assesses the proficiency of a classifier in identifying negative samples, thereby indicating its efficacy in identifying negative individuals. The method for calculating specificity is as follows:(16)$$Specificity=\frac{TN}{TN+FP}$$

The MCC metric quantifies the degree of association between diagnostic outcomes and true outcomes. In the context of balanced data, an increase in both the ACC and MCC scores indicates an improvement in the quality of predictions. However, in the case of unbalanced data, MCC serves as a more precise indicator of the predictor’s quality compared to ACC. The procedure for computing MCC is as follows:(17)$$MCC=\frac{TP\times TN-FP\times FN}{\sqrt{\left(TP+FP)\times (TP+FN)\times (TN+FP)\times (TN+FN\right)}}$$

F-Measure is the weighted average of precision (P) and recall (R). Precision refers to the proportion of actual positives among those classified as positive; recall is an indicator of coverage, that is, the number of cases classified as positive:(18)$$P=\frac{TP}{TP+FP}$$
(19)$$R=\frac{TP}{TP+FN}$$
(20)$$F=\frac{\left(\alpha^{2}+1\right)P*R}{\alpha^{2}(P+R)},\alpha =1$$

Next, the swarm intelligence optimization algorithms used in the feature selection experiment include BCCRIME, BRIME, BPSO, bMFO, bALO, BSSA, bMVO, and bCS. Table 4 shows the parameter values set for each algorithm, which are no different from the original values of the algorithm. Furthermore, owing to the characteristics of the feature selection technique, the algorithm’s dimensionality is consistently established as the number of features present in the dataset, while the algorithm’s population size is consistently established as 20. Two of the optimization algorithms among the set of eight, namely bCS and BMFO, are discretized based on the original algorithm. Algorithms such as BSSA and bALO have been referenced in the literature by other scholars. Furthermore, the experiment was conducted on a Windows Server 2008R2 operating system in order to ensure uniformity of conditions across all experiments. The fundamental components of this apparatus comprise an Intel Xeon (R) CPUE5-2660v3 (2.60 GHz) and 16 GB of RAM, in conjunction with Matlab2017b to execute the code.

#### 5.2.2. Microseismic and Blast Dataset Experiment

This section aimed to assess the efficacy of the BCCRIME-FKNN model in practical earthquake prediction by utilizing real-world microseismic and blasting data for classification prediction. This study aimed to evaluate the efficacy of the BCCRIME and FKNN combination by comparing it with three alternative predictors, namely KELM, KNN, and MLP. The comparison was conducted using microseismic and blasting datasets. The experiment aimed to demonstrate the performance disparity between BCCRIME-FKNN and conventional methods. To achieve this, a comparative analysis was conducted between BCCRIME and four classical methods, namely CART and RandomF, among others.

We executed all tests in this paper in accordance with fair comparison principles, aligned with rules employed in previous research [75,76]. A fair comparison involves using consistent metrics, datasets, and evaluation criteria throughout all methods or models being assessed to guarantee impartial results. Hence, the algorithmic configurations and code were sourced from the MATLAB defaults. Furthermore, to demonstrate the superiority of BCCRIME in comparison to other analogous algorithms, the present study conducted a comparative analysis between BCCRIME and seven other swarm intelligence algorithms, namely BRIME, BPSO, bMFO, bALO, BSSA, bMVO, and bCS. In order to comprehensively demonstrate the dependability of the prediction outcomes, this section employed four evaluation techniques, namely, accuracy, specificity, MCC, and F-value, to more precisely assess the processing capability of the BCCRIME-FKNN model on actual data.

Using various classifiers in conjunction with BCCRIME is likely to yield varying classification outcomes. In order to evaluate the efficacy of the BCCRIME and FKNN combination, the former was integrated with three additional classifiers and subjected to a comparative experiment. The findings are illustrated in Figure 7. The superior performance of BCCRIME-FKNN in terms of accuracy, specificity, MCC, and F-value is evident when compared to other predictors. Additionally, the stability of the FKNN model is discernible through the utilization of box plots. On the contrary, the amalgamation of BCCRIME with KELM and MLP exhibits inadequate precision and lacks robustness. Thus, it can be inferred that the amalgamation of BCCRIME and FKNN is highly appropriate.

This paper conducted a comparative experiment between BCCRIME-FKNN and classical classification methods to demonstrate the performance advantage of the former. Figure 8 displays the box plot outcomes of the conducted experiment. Evidently, BCCRIME-FKNN exhibited a substantial edge over alternative classification techniques. Within the set of algorithms considered, RandomF exhibited sporadic instances of favorable outcomes, yet its consistency was notably inferior to that of BCCRIME-FKNN. Conversely, the overall efficacies of CART and ELMforFS were generally subpar. The experimental findings indicate that BCCRIME-FKNN exhibited a significant advantage over traditional classification techniques.

The aforementioned experiments provide evidence that the amalgamation of FKNN and swarm intelligence algorithms yields superior performance. This study aimed to evaluate the efficacy of the BCCRIME-FKNN classification prediction model in comparison to other well-known swarm intelligence algorithms, such as BRIME, BPSO, bMFO, bALO, BSSA, bMVO, and bCS. The objective was to determine whether BCCRIME-FKNN outperforms these algorithms in terms of classification performance. A graphical representation of the comparison experiment’s outcomes is depicted in Figure 9 through the utilization of a box plot. The data from BCCRIME exhibited the highest values across all four classification evaluation indicators, suggesting that the BCCRIME-FKNN model possesses robust predictive capabilities relative to similar algorithms and is well-suited for forecasting microseismic and blasting issues. In contrast to the original MFO algorithm, ICCRIME-FKNN exhibited superior accuracy and enhanced stability. The BCCRIME model-based comprehensive classification effect exhibited superior accuracy and stability in relation to precision and error. The BCCRIME model-based selection exhibited superior accuracy for both negative and positive cases, particularly in terms of specificity. Regarding Matthew’s Correlation Coefficient (MCC), the BCCRIME model exhibited the highest correlation between the predicted and true values. The BCCRIME model exhibited a stronger comprehensive classification performance as indicated by its F-value.

In order to bolster the legitimacy of the comparative study, the present paper employed the Friedman test technique to authenticate and order the findings obtained from the experiment, which are presented in Table 5. Upon examination of the testing methodology, it can be concluded that BCCRIME-FKNN exhibited a high degree of stability, ranking first among the tested methods. Thus, based on the aforementioned experiments, it can be demonstrated that the BCCRIME-FKNN model is highly appropriate for aiding in the prediction of microseismic and blasting topics and is capable of efficiently categorizing microseismic and blasting datasets.

The experimental data from the CCI dataset based on the BCCRIME-FKNN model are presented in Table 6. The initial column denotes the label of ten-fold cross-validation, while the second column represents the count of selected features (SSFS). The remaining columns indicate the accuracy, sensitivity, MCC, and F-value employed in the aforementioned analysis. The results of the ten-fold cross-validation indicate an accuracy value of 90.91%, a specificity value of 0.9286, an MCC value of 0.8147, and an F-value of 0.8819. This finding demonstrates the efficacy of integrating BCCRIME and FKNN as a microseismic and blasting classifier model.

## 6. Conclusions and Future Directions

This paper proposed a RIME algorithm based on a vertical and horizontal cross-search strategy (CCRIME) to improve the quality of the solution obtained by the RIME algorithm and further enhance its search capability. By constructing a binary version of CCRIME, the key parameters in FKNN were optimized using binary conversion methods. In addition, this paper also proposed a BCCRIME based on discrete CCRIME, which converts the solution after searching into real numbers containing only 0 and 1 through the S-shaped function conversion method for solving feature selection problems. This paper studied the performance of CCRIME from multiple angles using 30 benchmark functions in IEEE CEC2017, mainly by conducting basic algorithm comparison experiments and advanced variant algorithm comparison experiments. Then, this paper also used collected microseismic and blasting data for classification prediction to verify the ability of the BCCRIME-FKNN model to handle real data, using four evaluation methods (the accuracy, specificity, MCC, and F-value) to display the reliability of the prediction results comprehensively, and it conducted ten-fold cross-validation on the experimental results. In summary, this paper proposed an effective method to solve the real-time rock mass stability monitoring problem during deep well mineral resource exploitation. By using the CCRIME algorithm combined with the FKNN model, this paper successfully identified microseismic and blasting signals and achieved high recognition accuracy. This research provides new ideas and methods for real-time monitoring of rock mass stability during deep well mineral resource exploitation.

In future research, it is possible to try to combine CCRIME with other optimization algorithms to improve its performance. It is also possible to explore the use of other evaluation methods to evaluate the reliability of the prediction results and conduct a more in-depth analysis of the experimental results.

## Figures and Tables

**Figure 1 sensors-23-08787-f001:**
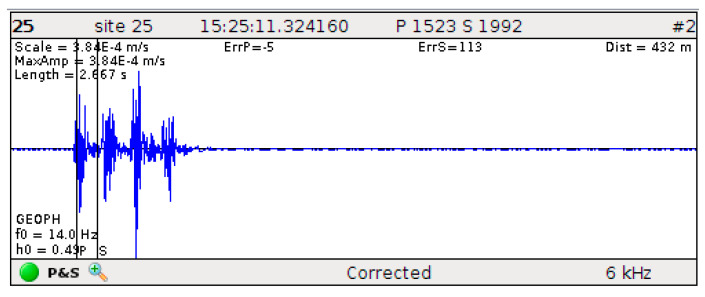
Monitoring and control interface of the Linglong Gold Mine microseismic system software.

**Figure 2 sensors-23-08787-f002:**
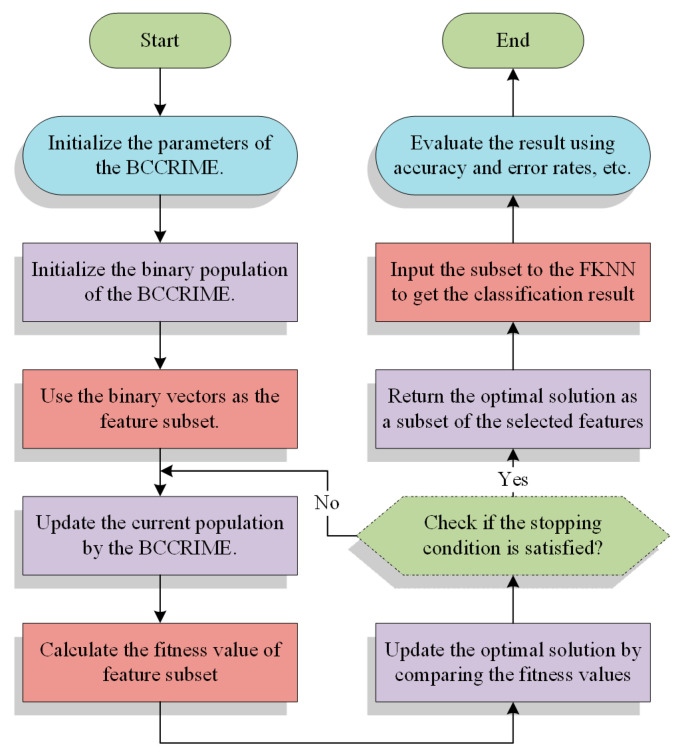
The flowchart of the developed CCRIME-FKNN model.

**Figure 3 sensors-23-08787-f003:**
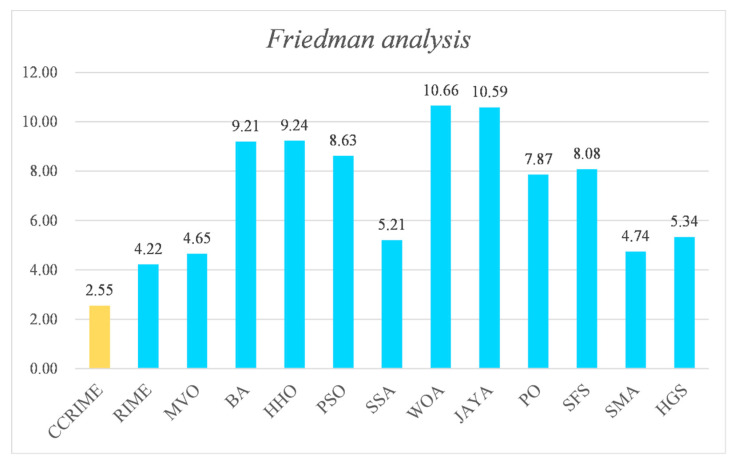
The ranking results after the Friedman test analysis.

**Figure 4 sensors-23-08787-f004:**
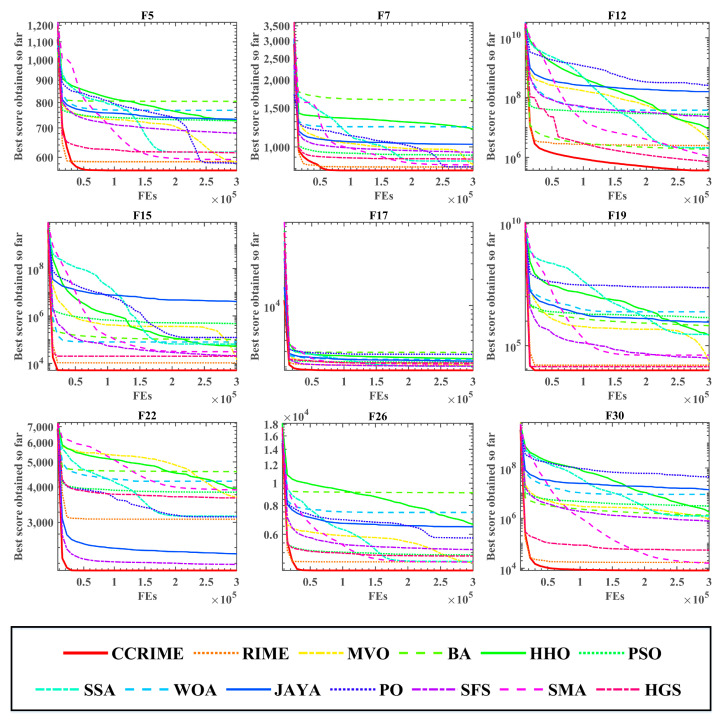
The convergence curves of some functions.

**Figure 5 sensors-23-08787-f005:**
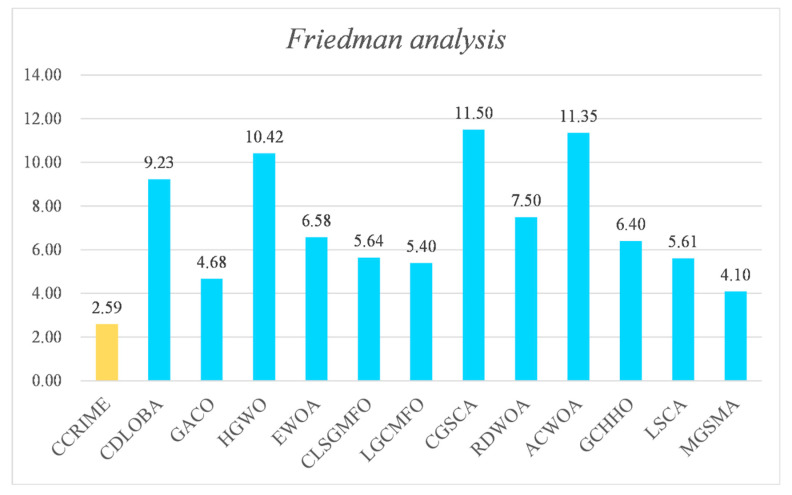
The Friedman ranking results of each algorithm.

**Figure 6 sensors-23-08787-f006:**
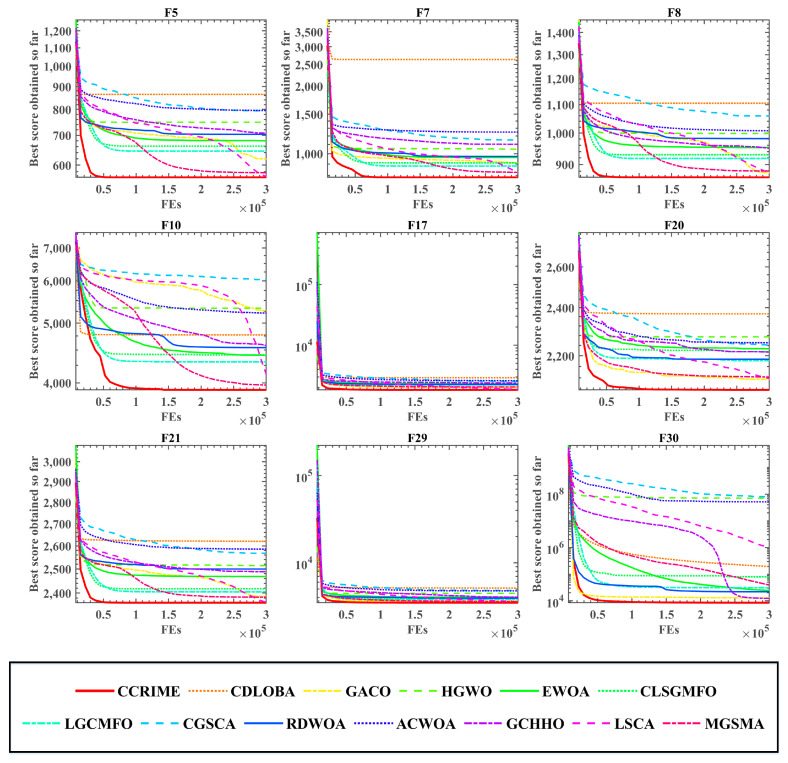
The convergence curves of all algorithms on some functions.

**Figure 7 sensors-23-08787-f007:**
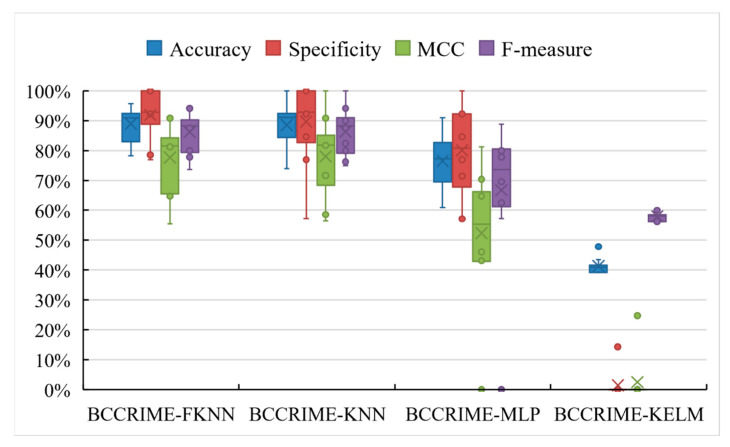
Results of five different classifiers based on BCCRIME.

**Figure 8 sensors-23-08787-f008:**
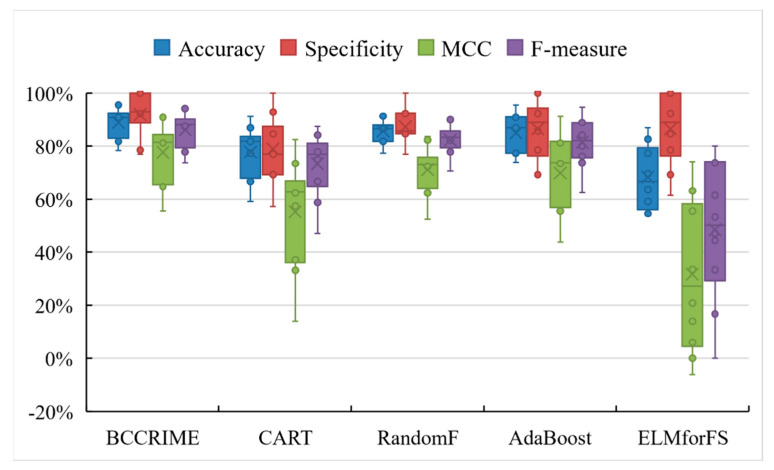
Comparison results of BCCRIME−FKNN with other classical methods.

**Figure 9 sensors-23-08787-f009:**
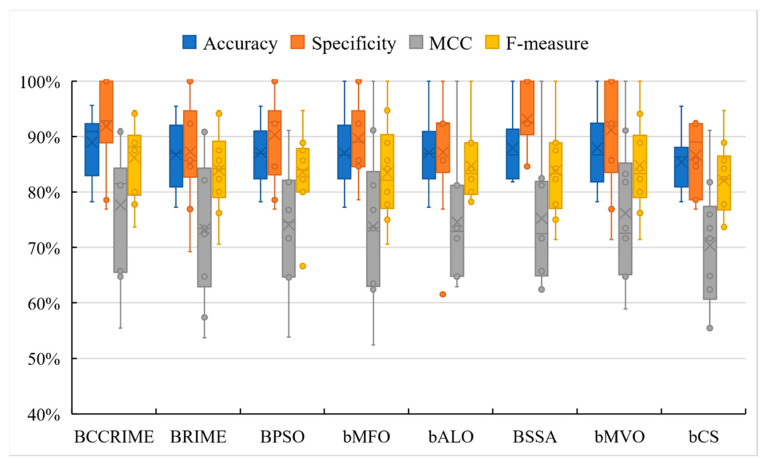
Comparison of BCCRIME and seven algorithms on four evaluation criteria.

**Table 1 sensors-23-08787-t001:** The description of 30 benchmark functions.

Item	The Function Class	The Function Name	Search Space	The Optimal Fitness
F1	Unimodal Functions	Shifted and Rotated Bent Cigar Function	[−100, 100]	100
F2		Shifted and Rotated Sum of Different Power Function	[−100, 100]	200
F3		Shifted and Rotated Zakharov Function	[−100, 100]	300
F4	Multimodal Functions	Shifted and Rotated Rosenbrocks Function	[−100, 100]	400
F5		Shifted and Rotated Rastrigins Function	[−100, 100]	500
F6		Shifted and Rotated Expanded Scaffers F6 Function	[−100, 100]	600
F7		Shifted and Rotated Lunacek Bi_Rastrigin Function	[−100, 100]	700
F8		Shifted and Rotated Non-Continuous Rastrigins Function	[−100, 100]	800
F9		Shifted and Rotated Levy Function	[−100, 100]	900
F10		Shifted and Rotated Schwefels Function	[−100, 100]	1000
F11	Hybrid Functions	Hybrid Function 1 (N = 3)	[−100, 100]	1100
F12		Hybrid Function 2 (N = 3)	[−100, 100]	1200
F13		Hybrid Function 3 (N = 3)	[−100, 100]	1300
F14		Hybrid Function 4 (N = 4)	[−100, 100]	1400
F15		Hybrid Function 5 (N = 4)	[−100, 100]	1500
F16		Hybrid Function 6 (N = 4)	[−100, 100]	1600
F17		Hybrid Function 6 (N = 5)	[−100, 100]	1700
F18		Hybrid Function 6 (N = 5)	[−100, 100]	1800
F19		Hybrid Function 6 (N = 5)	[−100, 100]	1900
F20		Hybrid Function 6 (N = 6)	[−100, 100]	2000
F21	Composition Functions	Composition Function 1 (N = 3)	[−100, 100]	2100
F22		Composition Function 2 (N = 3)	[−100, 100]	2200
F23		Composition Function 3 (N = 4)	[−100, 100]	2300
F24		Composition Function 4 (N = 4)	[−100, 100]	2400
F25		Composition Function 5 (N = 5)	[−100, 100]	2500
F26		Composition Function 6 (N = 5)	[−100, 100]	2600
F27		Composition Function 7 (N = 6)	[−100, 100]	2700
F28		Composition Function 8 (N = 6)	[−100, 100]	2800
F29		Composition Function 9 (N = 3)	[−100, 100]	2900
F30		Composition Function 10 (N = 3)	[−100, 100]	3000

**Table 2 sensors-23-08787-t002:** The performance ranking of all algorithms on each benchmark function.

	F1	F2	F3	F4	F5	F6	F7	F8	F9	F10	F11
CCRIME	2	4	6	4	1	1	1	1	1	3	1
RIME	5	7	5	8	4	2	2	2	2	1	2
MVO	6	6	4	7	3	5	5	4	3	5	7
BA	7	1	3	2	13	13	13	13	13	9	10
HHO	9	8	9	10	9	11	11	9	11	8	5
PSO	11	9	7	1	10	10	8	10	10	11	8
SSA	1	3	1	6	7	9	6	5	5	7	6
WOA	8	10	13	11	12	12	12	11	12	12	12
JAYA	13	12	11	13	11	7	10	12	6	13	13
PO	10	13	12	9	2	6	3	6	7	6	9
SFS	12	11	10	12	8	8	9	8	8	10	11
SMA	4	2	2	5	5	3	4	3	4	4	4
HGS	3	5	8	3	6	4	7	7	9	2	3
	F12	F13	F14	F15	F16	F17	F18	F19	F20	F21	F22
CCRIME	1	2	6	1	1	1	1	1	1	1	1
RIME	6	1	5	2	2	4	7	3	4	5	4
MVO	7	5	3	3	5	5	3	4	2	4	8
BA	4	8	2	10	12	13	4	9	13	13	13
HHO	8	9	8	7	9	11	10	7	9	10	11
PSO	10	11	4	12	8	8	5	11	10	11	9
SSA	5	6	1	8	4	2	2	8	3	6	6
WOA	11	7	13	9	13	10	12	12	12	12	12
JAYA	12	12	11	13	11	9	11	10	8	9	3
PO	13	13	12	11	10	12	13	13	11	2	5
SFS	9	10	10	5	6	3	9	5	5	8	2
SMA	3	4	7	6	3	6	8	6	6	3	10
HGS	2	3	9	4	7	7	6	2	7	7	7
	F23	F24	F25	F26	F27	F28	F29	F30	+/−/=	Mean	Rank
CCRIME	1	1	1	1	4	2	1	1	N/A	1.8	1
RIME	3	5	5	4	5	6	2	3	21/2/7	3.87	2
MVO	2	2	3	2	2	4	4	6	22/2/6	4.3	3
BA	13	12	9	13	13	1	13	8	25/4/1	9.33	9
HHO	12	13	10	11	11	9	9	9	29/0/1	9.43	11
PSO	11	10	8	7	1	8	8	10	25/1/4	8.57	8
SSA	5	3	7	5	7	3	6	7	23/3/4	5	5
WOA	10	9	11	12	9	10	12	11	30/0/0	11.07	13
JAYA	9	8	13	10	10	12	10	12	30/0/0	10.47	12
PO	8	11	6	9	12	13	11	13	27/0/3	9.37	10
SFS	7	7	12	8	8	11	7	5	30/0/0	8.13	7
SMA	4	4	2	3	3	7	3	2	23/2/5	4.33	4
HGS	6	6	4	6	6	5	5	4	24/0/6	5.33	6

**Table 3 sensors-23-08787-t003:** The ranking of all algorithms on each function.

	F1	F2	F3	F4	F5	F6	F7	F8	F9	F10	F11
CCRIME	2	1	2	2	1	1	1	1	1	1	1
CDLOBA	4	8	4	1	13	13	13	13	13	9	10
GACO	8	4	5	3	4	2	5	2	2	11	4
HGWO	12	12	13	11	10	9	9	10	6	12	13
EWOA	3	7	6	4	7	8	7	8	10	5	5
CLSGMFO	6	6	7	7	6	7	6	6	7	6	6
LGCMFO	7	5	9	5	5	5	4	5	5	4	7
CGSCA	13	13	11	13	11	11	11	12	11	13	11
RDWOA	9	9	10	9	8	6	8	9	9	7	8
ACWOA	11	11	12	12	12	12	12	11	12	10	12
GCHHO	1	3	3	8	9	10	10	7	8	8	9
LSCA	10	10	8	10	2	4	3	3	4	3	3
MGSMA	5	2	1	6	3	3	2	4	3	2	2
	F12	F13	F14	F15	F16	F17	F18	F19	F20	F21	F22
CCRIME	1	4	3	3	2	1	2	1	1	1	1
CDLOBA	3	8	1	10	10	13	1	9	13	13	13
GACO	2	5	6	8	4	3	8	8	2	4	12
HGWO	11	12	13	13	11	10	11	11	12	10	4
EWOA	7	3	9	7	6	8	9	5	9	7	9
CLSGMFO	4	9	8	4	7	6	5	4	8	6	2
LGCMFO	5	6	4	1	5	5	3	2	5	5	3
CGSCA	13	13	11	12	12	11	13	13	10	11	5
RDWOA	9	2	10	5	9	7	10	6	6	9	11
ACWOA	12	11	12	11	13	12	12	12	11	12	8
GCHHO	6	1	5	2	8	9	6	3	7	8	6
LSCA	10	10	7	9	1	2	7	10	3	2	10
MGSMA	8	7	2	6	3	4	4	7	4	3	7
	F23	F24	F25	F26	F27	F28	F29	F30	+/−/=	Mean	Rank
CCRIME	2	2	3	3	3	1	1	1	N/A	1.67	1
CDLOBA	13	13	10	13	13	7	13	9	24/1/5	9.53	10
GACO	3	6	1	6	4	2	4	3	20/0/10	4.7	3
HGWO	9	8	11	10	10	11	10	12	30/0/0	10.53	11
EWOA	7	7	7	7	6	6	8	5	26/0/4	6.73	8
CLSGMFO	6	5	4	2	9	8	6	8	24/0/6	6.03	6
LGCMFO	5	4	5	1	8	4	5	6	25/0/5	4.77	4
CGSCA	11	10	13	11	11	13	12	13	30/0/0	11.6	13
RDWOA	8	11	8	9	5	10	7	4	28/0/2	7.93	9
ACWOA	12	12	12	12	12	12	11	11	30/0/0	11.57	12
GCHHO	10	9	6	8	7	3	9	2	24/1/5	6.37	7
LSCA	1	1	9	5	2	9	2	10	22/1/7	5.67	5
MGSMA	4	3	2	4	1	5	3	7	21/3/6	3.9	2

**Table 4 sensors-23-08787-t004:** Parameter settings for the optimization algorithms.

Algorithms	BCCRIME	bMFO	BSSA	bMFO
Values	W = 5	W = 5	~	a = 2; b = 1
Algorithms	bALO	bMVO	BPSO	bCS
Values	~	Max = 1; Min = 0.2	wMax = 0.9; wMin = 0.2	pa = 0.25

**Table 5 sensors-23-08787-t005:** Verification of the experimental results by using the Friedman test method.

Method		BCCRIME	BRIME	BPSO	bMFO	bALO	BSSA	bMVO	bCS
Accuracy	Avg	3.65	4.6	4.5	4.4	4.65	4.45	4.4	5.35
	Rank	**1**	6	5	2	7	4	2	8
Specificity	Avg	3.55	4.8	4.35	4.6	5	3.75	4.1	5.85
	Rank	**1**	6	4	5	7	2	3	8
MCC	Avg	3.75	4.6	4.55	4.2	4.7	4.6	4.4	5.2
	Rank	**1**	5	4	2	7	5	3	8
F-measure	Avg	3.5	4.5	4.5	4.55	4.55	4.65	4.6	5.15
	Rank	**1**	2	2	4	4	7	6	8

**Table 6 sensors-23-08787-t006:** Detailed results obtained by BCCRIME.

Fold	SSFS	Accuracy	Specificity	MCC	F-Measure
#1	91	0.909	0.923	0.812	0.889
#2	81	0.909	1.000	0.821	0.875
#3	83	0.955	1.000	0.909	0.941
#4	57	0.909	0.923	0.812	0.889
#5	82	0.957	0.929	0.914	0.947
#6	89	0.818	0.769	0.647	0.800
#7	79	0.833	0.929	0.657	0.778
#8	63	0.913	0.929	0.818	0.889
#9	87	0.783	0.786	0.555	0.737
#10	80	0.909	1.000	0.821	0.875
AVG	~	**0.909**	**0.929**	**0.815**	**0.882**
STD	~	**0.058**	**0.082**	**0.118**	**0.069**

## Data Availability

The data involved in this study are all public data, which can be downloaded through public channels.

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
