# Peer review of "An Enhanced RIME Optimizer with Horizontal and Vertical Crossover for Discriminating Microseismic and Blasting Signals in Deep Mines"

_sensors, 2023, doi:10.3390/s23218787_

Round 1
Reviewer 1 Report
The topic of this paper is interesting, but there are some issues that need to be addressed before publication.
1. Please explain how the input and output data are collected, pre-processed, analysed and used for training, testing and validation of the learning algorithm.
2. How to reflect the fairness of comparison between the RIME algorithm and existing methods?
3. In the introduction section, the author should cite recently developed algorithms such as:
https://doi.org/10.1016/j.knosys.2023.110679
4. The paper does not discuss how the proposed technique can be applied to different types of optimization problems.
Author Response
Comment 1:
Please explain how the input and output data are collected, pre-processed, analysed and used for training, testing and validation of the learning algorithm.
- Thanks for your valuable comment and kind suggestion; you have raised an important point. We have added and illustrated it in section 5.2.1 and marked the relevant content in blue.
Comment 2:
How to reflect the fairness of comparison between the RIME algorithm and existing methods?
- Thanks for your valuable comment and kind suggestion; you have raised an important point. We have added and illustrated it in section 5.1.1 and marked the relevant content in blue.
Comment 3:
In the introduction section, the author should cite recently developed algorithms such as: https://doi.org/10.1016/j.knosys.2023.110679
- Thanks for your valuable comment and kind suggestion; you have raised an important point. We have cited these newly developed algorithms in the introduction and labeled them blue.
Comment 4:
The paper does not discuss how the proposed technique can be applied to different types of optimization problems.
- Thanks for your valuable comment and kind suggestion; you have raised an important point. We have added the relevant content in section 6 and marked it blue in the new manuscript.
Reviewer 2 Report
Dear Authors,
Greetings. Many thanks for your manuscript submission to MDPI Jour. of Sensors. I would recommend this paper as "acceptance with minor revisions" given that the following aspects got significantly improved in your edits. There are some potentiality and room for improvement in this research article, despite some part of the current version is still a bit rough. The suggested improvements and comments for edits are listed as follows:
1) This paper proposed a smart optimization approach, i.e., RIME algorithm to improve the upcoming quality of solutions and further enhance its crossover searching capabilities. The authors adopted binary conversion method to optimize its crucial parameters and derived a discrete scheme on the variations of RIME (named as BCC-RIME) for binary feature selection. Basic information retrieval (IR) valuation metrics and 10-fold cross-validation tests are applied to verify the capacity of proposed model when processing collected micro-seismic and blasting data from IEEE CEC2017. Good job.
2) This paper presents a complete set of research study on the cross areas of machine learning, data mining and smart optimization algorithms, and have claimed on proposing new ideas and novel methods on real-time monitoring of rock mass stability when exploiting deep well mineral resources. Topic is intriguing, while actual innovations must be addressed.
3) Abstract: Do please shorten the beginning ten lines to make it more coherent and cohesive. Update the last two sentences with keynote quantitative scores in concluding remarks. Restrict the length of words to 180~220 in total. PS: This version fails to show any numbered lines, hence, I have to specify the actual page number and pointer of subsections.
4) Introduction: Fine but a few problematic issues co-exist in this version:
i) Please consider condensing the first and second paragraph when narrating historical review, and the problem solving on micro-seismic monitoring needs a major rewrite.
ii) Besides, Refs. [1] to [5] were applied with a larger font size, which must be 2-font smaller (starting from [6], they return to normal).
iii) Pages 3-4: It is very nice to summarize main research and contributions of this paper, while each short paragraph of the manifolds should be a bit more specific. Meanwhile, remove the unnecessary italic style of characters.
5) Section 2: This section is better to be "Related Work", for discussing a bit more state-of-the art models, keynote framework on handling the source of data, and description of data can be shrinked to a subsection. In addition, the content of Fig. 1 is hardly readable, both the image resolution and visual display of characters must be enhanced, along with the size of this figure.
6) Section 3: In Pages 5-7, please apply right alignment for each of the ten numbered equations first, and adjust the linspacing of each equations (which should be applied with middle alignment). Last paragraph of subsection 3.1, please shift "where" to the very beginning of this line (below Eq. (7)). PS: The summary of Algorithm 1, please italize "t" and "T" at the fifth line. Also, I think the summary of this algorithm should be a bit more specific, please leave one empty line between the context "The pseudocode is shown in Algorithm 1." and the front of this tablulated summary of Algorithm.
7) Pages 7-12: Please fix the similar problems in Section 4 and Section 5 first (i.e., Equations (11)-(14), and middle alignment for the content of Table 1). If possible, simplify the flowchart at Fig. 2 (shorten each element of the developed CCRIME-FKNN model). Thanks a lot!
PS: For machine learning based apporach, what are the evaluation metrics in addition to basic IR ones, i.e., loss function, mean average precision, etc.?
8) Tables 2-5 crossed over at least two pages (the longest Table 4 went through Pages 19-23, five pages in total), which are unacceptable. I suggest the authors transpose the horizontal and vertical part of corresponding elements and hence compress the height of very long tables (especially Table 4), Tables 3 and 5 can be re-arranged for covering one of each entire page, while Tables 2 and 4 must be adjusted to shorter Tables (denoted with Table X' continued), and round the numeric values to 2 valid digits after decimals.
9) Pages 25-26: Shift the title of "Figure 6 ..." to the bottom of last page. Apply the designated font style to "5.2. Feature selection experiments".
10) Pages 26-27: Align the position of each equation first, then check if the definition for Specificity is correct: https://zhuanlan.zhihu.com/p/479173794. If not, apply the required edits on Eq. (16). Direct specify harmonic measure of P and R in Eq. (20) is also quite OK. PS: replace "&" as "and" in the sub-title of subsection 5.2.2 and all other occurrences.
11) Pages 29-31: Apply middle-alignment for the digital numbers at Table 7, and I think 3 valid digits after each decimal for the number at Table 8, would be accurate enough. The paragraph for future directions is too simple and a bit too generic. The standard professional one should include opening questions, potential challenging problems to be solved, the suggested topics or a varitey of proposed orientations under further investigation. Please expand the second paragraph of Section 2, at least making it appeared as balanced as the first paragraph (in summary).
12) References: Suggestions for edits are in the following manifolds:
i) Apply the decent font style of this section first (and remove all capital letters at Ref. [4]). It is not necessary to be all italic for the name of research articles.
ii) Apply abbreviated style for required names of journals and conference proceedings.
iii) Make up the missed information (i.e., time, location, volumn, number, the start and end of page numbers) for Refs. [18], [24], [26], and [39].
iv) Besides, it is not compulsory to only cite references in the latest five years. I recommend the authors select a few classical but representative references in typical machine learning algorithms and intelligent optimzation strategy related approaches; if more latest and reliable schemes for RIME based methods, please include and update the list of references.
Minor revisions suggested for edits are as follows (not limited to these):
a) Please stop minor hypherating issues when crossing over two adjacent lines. Check an MS word version of MDPI template to resolve the issue.
b) Be uniform on the bold, italic, and standart font style of characters when denoting variables or formulas.
c) Check minor grammatical mistakes and awkward use of English phrases when proofreading the entire mauscript along with enhancing literal quality.
Above all, thank you very much for your interests on publishing at Sensors Journal, we expect you the best of luck for updated paper coming into the double decision process. Stay well!
With warm regards,
Yours sincerely,
In most part of this paper, the quality of literal writing is acceptable; while some sentences are still hardly readable, which needs further edits in English. I suggest one or two of the peer-reviewed authors preparing one round of proofreading to ensure the literal quality meets the critera of publication.
Author Response
Comment 1:
This paper proposed a smart optimization approach, i.e., RIME algorithm to improve the upcoming quality of solutions and further enhance its crossover searching capabilities. The authors adopted binary conversion method to optimize its crucial parameters and derived a discrete scheme on the variations of RIME (named as BCC-RIME) for binary feature selection. Basic information retrieval (IR) valuation metrics and 10-fold cross-validation tests are applied to verify the capacity of proposed model when processing collected micro-seismic and blasting data from IEEE CEC2017. Good job.
- Thanks for your acceptance and recognition of this work now, and for your highly constructive comments that helped us to further enhance the excellence of this research.
Comment 2:
This paper presents a complete set of research study on the cross areas of machine learning, data mining and smart optimization algorithms, and have claimed on proposing new ideas and novel methods on real-time monitoring of rock mass stability when exploiting deep well mineral resources. Topic is intriguing, while actual innovations must be addressed.
- Thanks for your acceptance and recognition of this work now, and for your highly constructive comments that helped us to further enhance the excellence of this research. In addition, in the introduction we have also redescribed the main contributions and innovations of this paper and marked them in blue.
Comment 3:
Abstract: Do please shorten the beginning ten lines to make it more coherent and cohesive. Update the last two sentences with keynote quantitative scores in concluding remarks. Restrict the length of words to 180~220 in total. PS: This version fails to show any numbered lines, hence, I have to specify the actual page number and pointer of subsections.
- Thanks for your valuable comment and kind suggestion; you have raised an important point. We have revised and adjusted the content and marked it blue in the new manuscript.
Comment 4:
Introduction: Fine but a few problematic issues co-exist in this version:
- i) Please consider condensing the first and second paragraph when narrating historical review, and the problem solving on micro-seismic monitoring needs a major rewrite.
- ii) Besides, Refs. [1] to [5] were applied with a larger font size, which must be 2-font smaller (starting from [6], they return to normal).
iii) Pages 3-4: It is very nice to summarize main research and contributions of this paper, while each short paragraph of the manifolds should be a bit more specific. Meanwhile, remove the unnecessary italic style of characters.
- Thanks for your valuable comment and kind suggestion; you have raised an important point. We have revised and adjusted the content and marked it blue in the new manuscript.
Comment 5:
Section 2: This section is better to be "Related Work", for discussing a bit more state-of-the art models, keynote framework on handling the source of data, and description of data can be shrinked to a subsection. In addition, the content of Fig. 1 is hardly readable, both the image resolution and visual display of characters must be enhanced, along with the size of this figure.
- Thanks for your valuable comment and kind suggestion; you have raised an important point. We have revised and adjusted the content and marked it blue in the new manuscript.
Comment 6:
Section 3: In Pages 5-7, please apply right alignment for each of the ten numbered equations first, and adjust the linspacing of each equations (which should be applied with middle alignment). Last paragraph of subsection 3.1, please shift "where" to the very beginning of this line (below Eq. (7)). PS: The summary of Algorithm 1, please italize "t" and "T" at the fifth line. Also, I think the summary of this algorithm should be a bit more specific, please leave one empty line between the context "The pseudocode is shown in Algorithm 1." and the front of this tablulated summary of Algorithm.
- Thanks for your valuable comment and kind suggestion; you have raised an important point. We have revised and adjusted the content and marked it blue in the new manuscript.
Comment 7:
Pages 7-12: Please fix the similar problems in Section 4 and Section 5 first (i.e., Equations (11)-(14), and middle alignment for the content of Table 1). If possible, simplify the flowchart at Fig. 2 (shorten each element of the developed CCRIME-FKNN model). Thanks a lot!
PS: For machine learning based apporach, what are the evaluation metrics in addition to basic IR ones, i.e., loss function, mean average precision, etc.?
- Thanks for your valuable comment and kind suggestion; you have raised an important point. We have revised and adjusted the content and marked it blue in the new manuscript.
Comment 8:
Tables 2-5 crossed over at least two pages (the longest Table 4 went through Pages 19-23, five pages in total), which are unacceptable. I suggest the authors transpose the horizontal and vertical part of corresponding elements and hence compress the height of very long tables (especially Table 4), Tables 3 and 5 can be re-arranged for covering one of each entire page, while Tables 2 and 4 must be adjusted to shorter Tables (denoted with Table X' continued), and round the numeric values to 2 valid digits after decimals.
- Thanks for your valuable comment and kind suggestion; you have raised an important point. We have revised and adjusted the content and marked it blue in the new manuscript. In addition, we have moved some lengthy tables to the appendix.
Comment 9:
Pages 25-26: Shift the title of "Figure 6 ..." to the bottom of last page. Apply the designated font style to "5.2. Feature selection experiments".
- Thanks for your valuable comment and kind suggestion; you have raised an important point. We have revised and adjusted the relevant content in the new manuscript.
Comment 10:
Pages 26-27: Align the position of each equation first, then check if the definition for Specificity is correct: https://zhuanlan.zhihu.com/p/479173794. If not, apply the required edits on Eq. (16). Direct specify harmonic measure of P and R in Eq. (20) is also quite OK. PS: replace "&" as "and" in the sub-title of subsection 5.2.2 and all other occurrences.
- Thanks for your valuable comment and kind suggestion; you have raised an important point. We have checked and revised the content and marked it blue in the new manuscript.
Comment 11:
Pages 29-31: Apply middle-alignment for the digital numbers at Table 7, and I think 3 valid digits after each decimal for the number at Table 8, would be accurate enough. The paragraph for future directions is too simple and a bit too generic. The standard professional one should include opening questions, potential challenging problems to be solved, the suggested topics or a varitey of proposed orientations under further investigation. Please expand the second paragraph of Section 2, at least making it appeared as balanced as the first paragraph (in summary).
- Thanks for your valuable comment and kind suggestion; you have raised an important point. We have revised and adjusted the content and marked it blue in the new manuscript.
Comment 12:
References: Suggestions for edits are in the following manifolds:
- i) Apply the decent font style of this section first (and remove all capital letters at Ref. [4]). It is not necessary to be all italic for the name of research articles.
- ii) Apply abbreviated style for required names of journals and conference proceedings.
iii) Make up the missed information (i.e., time, location, volumn, number, the start and end of page numbers) for Refs. [18], [24], [26], and [39].
- iv) Besides, it is not compulsory to only cite references in the latest five years. I recommend the authors select a few classical but representative references in typical machine learning algorithms and intelligent optimzation strategy related approaches; if more latest and reliable schemes for RIME based methods, please include and update the list of references.
- Thanks for your valuable comment and kind suggestion; you have raised an important point. We have revised and adjusted the content and marked it blue in the new manuscript.
Comment 13:
Please stop minor hypherating issues when crossing over two adjacent lines. Check an MS word version of MDPI template to resolve the issue.
- Thanks for your valuable comment and kind suggestion; you have raised an important point. We have modified and adjusted the content and have adopted the MDPI template.
Comment 14:
Be uniform on the bold, italic, and standart font style of characters when denoting variables or formulas.
- Thanks for your valuable comment and kind suggestion; you have raised an important point. We have checked the full article and revised the relevant content.
Comment 15:
Check minor grammatical mistakes and awkward use of English phrases when proofreading the entire mauscript along with enhancing literal quality.
- Thanks for your valuable comment and kind suggestion; you have raised an important point. We have rewritten some phrases and sentences in the manuscript, and it also has been polished by an English linguist. Thank you again for your valuable suggestions.
Author Response
Comment 1:
The abstract is too long and technical, and could be simplified and shortened to highlight the main contributions and findings of the paper.
- Thanks for your valuable comment and kind suggestion; you have raised an important point. We have revised and adjusted the content and marked it blue in the new manuscript.
Comment 2:
Define some key terms and abbreviations that are used throughout the paper.
- Thanks for your valuable comment and kind suggestion; you have raised an important point. We have defined some key terms and abbreviations that are used throughout the paper.
Comment 3:
The introduction of the manuscript could be further enhanced by providing additional context regarding how this research builds upon or diverges from prior work within the same field, e.g., (https://doi.org/10.1016/j.compgeo.2023.105692), (https://doi.org/10.1155/2017/6059239), (https://doi.org/10.3390/fractalfract6070361 ).
- Thanks for your valuable comment and kind suggestion; you have raised an important point. We have added and adjusted the relevant content in the introduction and marked it blue in the new manuscript.
Comment 4:
The data description section could provide more details on how the data was collected, processed, and labeled, as well as some descriptive statistics or visualizations of the data.
- Thanks for your valuable comment and kind suggestion; you have raised an important point. We have revised and adjusted the content in the section 2.1 and marked it blue in the new manuscript.
Comment 5:
The proposed CCRIME section could explain more clearly the rationale and intuition behind the algorithm, as well as its advantages over other swarm intelligence algorithms.
- Thanks for your valuable comment and kind suggestion; you have raised an important point. We have explained and added the relevant content in subsection 3.3 and marked it blue in the new manuscript.
Comment 6:
Please include line numbers alongside the content of the article to facilitate the review process.
- Thanks for your valuable comment and kind suggestion; you have raised an important point. We have taken the MDPI template and added line numbers.
Comment 7:
One critical issue that needs attention is the lack of integration between the text and the supplementary materials, including images, tables, equations, and references. Throughout the manuscript, it is essential to ensure that these elements are properly linked to the relevant sections of the text. This will enhance the overall clarity and readability of the paper and help readers to easily locate and understand the supporting materials that complement your findings.
- Thanks for your valuable comment and kind suggestion; you have raised an important point. We have reorganized the supplementary material and the main article.
Comment 8:
The conclusion and future work section could summarize the main contributions and findings of the paper, and suggest some directions for further research or improvement.
- Thanks for your valuable comment and kind suggestion; you have raised an important point. We have revised and adjusted the content and marked it blue in the new manuscript.